# Influenza Vaccine Effectiveness against Hospitalization, Season 2021/22: A Test-Negative Design Study in Barcelona

**DOI:** 10.3390/vaccines11091450

**Published:** 2023-09-02

**Authors:** Mar Fornaguera, Oleguer Parés-Badell, Íngrid Carbonés-Fargas, Cristina Andrés, José Ángel Rodrigo-Pendás, Blanca Borras-Bermejo, Lluís Armadans-Gil, Gabriela Tejada, David Guananga, Martí Vivet-Escalé, Arnau Peñalver-Piñol, Irene Torrecilla-Martínez, Arnau del Oso, Xavier Martínez-Gómez, Andres Antón, Susana Otero-Romero

**Affiliations:** 1Servei de Medicina Preventiva i Epidemiologia, Vall d′Hebron Hospital Universitari, Vall d′Hebron Barcelona Hospital Campus, Passeig Vall d′Hebron 119-129, 08035 Barcelona, Spain; mar.fornaguera@vallhebron.cat (M.F.); josean.rodrigo@vallhebron.cat (J.Á.R.-P.); blanca.borras@vallhebron.cat (B.B.-B.); lluis.armadans@vallhebron.cat (L.A.-G.); gabriela.tejada@vallhebron.cat (G.T.); david.guananga@vallhebron.cat (D.G.); marti.vivet@vallhebron.cat (M.V.-E.); arnau.penalver@vallhebron.cat (A.P.-P.); itorrecillam.germanstrias@gencat.cat (I.T.-M.); xavier.martinez@vallhebron.cat (X.M.-G.); susana.otero@vallhebron.cat (S.O.-R.); 2Grup de recerca de Epidemiologia i Salut Pública, Vall d′Hebron Institut de Recerca (VHIR), Vall d′Hebron Hospital Universitari, Vall d′Hebron Barcelona Hospital Campus, Passeig Vall d′Hebron 119-129, 08035 Barcelona, Spain; ingrid.carbones@vhir.org; 3Unitat Docent Vall d′Hebron, Universitat Autònoma de Barcelona, Passeig Vall d′Hebron 119-129, 08035 Barcelona, Spain; 4Respiratory Virus Unit, Microbiology Department, Vall d′Hebron Research Institute, Vall d′Hebron Hospital Universitari, Vall d′Hebron Barcelona Hospital Campus, Passeig Vall d′Hebron 119-129, 08035 Barcleona, Spain; cristina.andresverges@vallhebron.cat (C.A.); andres.anton@vallhebron.cat (A.A.); 5CIBERINFEC, ISCIII-CIBER de Enfermedades Infecciosas, Instituto de Salud Carlos III, Av. de Monforte de Lemos, 5, 28029 Madrid, Spain; 6Department Information Systems and Decision Support, Vall d′Hebron Barcelona Hospital Campus, Passeig Vall d′Hebron 119-129, 08035 Barcelona, Spain; arnau.deloso@vallhebron.cat

**Keywords:** influenza, vaccine, effectiveness, hospitalization

## Abstract

Background: Vaccination is considered the most effective measure for preventing influenza and its complications. The influenza vaccine effectiveness (IVE) varies annually due to the evolution of influenza viruses and the update of vaccine composition. Assessing the IVE is crucial to facilitate decision making in public health policies. Aim: to estimate the IVE against hospitalization and its determinants in the 2021/22 season in a Spanish tertiary hospital. Methods: We conducted a prospective observational test-negative design study within the Development of Robust and Innovative Vaccine Effectiveness (DRIVE) project. Hospitalized patients with severe acute respiratory infection (SARI) and an available influenza reverse transcription polymerase chain reaction (RT-PCR) were selected and classified as cases (positive influenza RT-PCR) or controls (negative influenza RT-PCR). Vaccine information was obtained from electronic clinical records shared by public healthcare providers. Information about potential confounders was obtained from hospital clinical registries. The IVE was calculated by subtracting the ratio of the odds of vaccination in cases and controls from one, as a percentage (IVE = (1 − odds ratio (OR)) × 100). Multivariate IVE estimates were calculated using logistic regression. Results: In total, 260 severe acute respiratory infections (SARI) were identified, of which 34 were positive for influenza, and all were subtype A(H3N2). Fifty-three percent were vaccinated. Adjusted IVE against hospitalization was 26.4% (95% CI −69% to 112%). IVE determinants could not be explored due to sample size limitations. Conclusion: Our data revealed non-significant moderate vaccine effectiveness against hospitalization for the 2021/2022 season.

## 1. Introduction

Seasonal influenza viruses are responsible for an estimated range of 4–50 million documented symptomatic infections annually in the European Union (EU), resulting in 15,000–70,000 deaths [1]. Vaccination is considered the most effective measure for preventing influenza and its complications [2]. Due to frequent genetic and antigenic changes in influenza viruses, an annual reformulation of the influenza vaccine is undertaken in accordance with guidance from the World Health Organization (WHO). For the 2021/22 northern hemisphere influenza season, the influenza vaccine composition was A/Victoria/2570/2019 (H1N1)pdm09-like virus, A/Cambodia/e0826360/2020 (H3N2)-like virus, B/Washington/02/2019 (B/Victoria lineage)-like virus, and B/Phuket/3073/2013 (B/Yamagata lineage)-like virus for egg-based vaccines; and A/Wisconsin/588/2019 (H1N1)pdm09-like virus, A/Cambodia/e0826360/2020 (H3N2)-like virus, B/Washington/02/2019 (B/Victoria lineage)-like virus, and B/Phuket/3073/2013 (B/Yamagata lineage)-like virus for cell- or recombinant-based vaccines [3].

According to the data collected by the European Centre for Disease Prevention and Control (ECDC) across the WHO European Region, the 2021/22 influenza season was characterized by a low incidence of cases, long duration, predominance of influenza A(H3N2), and sporadic circulation of influenza A (H1N1)pdm09 and B/Victoria viruses [4]. Similar characteristics of influenza virus circulation have been observed in Spain, with two recorded epidemic waves and a lower hospital burden than in the three seasons prior to the COVID-19 pandemic [5].

Influenza vaccine effectiveness (IVE) fluctuates due to multiple factors. These include characteristics of the recipient (age and health status), of the vaccine (components, types, subtypes (for influenza A) and linages (for influenza B included)), of the circulating influenza virus (virulence and transmission), and the similarity between circulating viruses and those included in the vaccine [6]. An extensive systematic review and meta-analysis by the ECDC illustrates this variability: trivalent IVE across three influenza seasons was estimated to be 45% (95% CI: 23% to 61%) against infection for any influenza virus type, but 61% (95% CI: 44% to 73%) for influenza A (H1N1)pdm09, 29% (95% CI: 5% to 46%) for influenza B, and 11% (95% CI: −25% to 36%) for influenza A(H3N2) [7].

Under requirements of the European Medicines Agency, seasonal and pandemic IVE has been monitored since the 2008/09 influenza season through the Influenza Monitoring Vaccine Effectiveness (I-MOVE) project [8]. In Spain, influenza surveillance is carried out through the “Sistema de Vigilancia de la Gripe y otros virus respiratorios en España (SVGE)”, based on the detection of mild and severe acute respiratory infections (SARI) from physicians and sentinel centers [9]. In 2017, the Development of Robust and Innovative Vaccine Effectiveness (DRIVE) consortium was formed with the aim to create a European platform for studying brand-specific IVE and to develop a governance model for scientifically robust, independent, and transparent implementation of IVE studies in a public–private partnership [10]. Hospital Universitari Vall d’Hebron joined the DRIVE project in 2018.

In the present study based on the DRIVE project, we aim to estimate the IVE for preventing hospitalization due to influenza and its determinants during the season 2021/22 in Hospital Universitari Vall d’Hebron.

## 2. Methods

### 2.1. Study Design and Population

We conducted a test-negative design (TND) case-control study based on the DRIVE protocol [11]. Figure 1 provides a summarized representation of the methodological design in the form of a flowchart. It was conducted in Hospital Universitari Vall d’Hebron, a tertiary hospital which provides healthcare coverage to 430,000 inhabitants of Catalonia and is an international referent for highly complex tertiary procedures. The study was conducted during the 2021/22 influenza season, from the 29 October 2021 until the 20 April 2022.

The study population consisted of all patients admitted to the hospital with SARI according to the WHO accepted definition; this being an acute respiratory infection with a history of fever and cough with onset within the last 10 days that requires hospitalization [12]. Following routine procedures, respiratory specimens were collected from all hospitalized SARI patients and tested for influenza using a reverse transcription polymerase chain reaction (RT-PCR). The inclusion criteria were: being hospitalized for SARI and having a diagnostic test for influenza performed during the admission. The exclusion criteria were: being institutionalized, an age younger than 6 months, contraindication for influenza vaccination, a prior positive influenza test in the same season, a swab taken more than 8 days after SARI onset, and an influenza vaccination less than 15 days before SARI onset. Patients were classified into cases and controls according to the positive or negative swab test result, respectively.

Based on the hospital’s electronic clinical records, hospitalized patients with signs or symptoms at admission suggestive of SARI were automatically selected. Subsequently, the admission reports were manually reviewed to confirm fulfilment of the inclusion criteria.

### 2.2. Sample Size

To calculate the sample size, we used the World Health Organization sample size calculator [13]. We calculated the sample size using the CIs values obtained from the ECDC meta-analysis, 44% to 66% [7], a precision of 40%, and an expected vaccination coverage rate of 50% [14]. Therefore, our sample size calculation ranged from 115 to 262 cases and from 115 to 262 controls.

### 2.3. Data and Sources of Information

For each patient, demographic information (age and sex), risk factors (smoking habits and comorbidities), as well as laboratory and vaccination data, were systematically collected from the Hospital Universitari Vall d’Hebron electonic clinical records.

Smoking habits were categorized into 4 groups: never smokers, ex-smokers, occasional smokers, and daily smokers. The following selected comorbidities were collected as categorical dichotomous variables according to ICD-10 diagnostic codes: cardiovascular disease, lung disease, renal disease, liver disease, diabetes, cancer, and immunodeficiency. The variable “any chronic condition” was created by taking into account all comorbidities. Laboratory data included influenza type (A and B) and subtype/lineage. Vaccination variables included date of vaccination, type of vaccine, previous vaccination for influenza (only season 20/21), and COVID-19 and pneumococcal vaccination record (any time). They were obtained from Història Clínica Compartida de Catalunya (HC3), an electronic clinical records system shared by all public healthcare centers in Catalonia. The variable COVID-19 infection history was defined as RT-PCR or antigen positive SARS-CoV-2 tests in the past 12 months that had been registered in HC3. We did not have access to serological SARS-CoV-2 tests.

The extraction of variables available in the automated hospital records was automatized except for the date of the onset of symptoms, smoking habits, and contraindication for influenza vaccine, which were manually retrieved.

Potential confounders or effect modifiers were selected based on population-based vaccine effectiveness studies available in the literature.

Patients with damaged samples, pending laboratory results, or missing data regarding the laboratory result outcomes or covariates were excluded from the analysis.

## 3. Ethical Statement

Databases were developed according to national and international standards of ethical practices (Declaration of Helsinki and Tokyo), and data were used in accordance with the regulations in force regarding the protection of personal data (EU) 2016/679; 27 April of 2016 (GDPR). The present study was approved by the Human Research Ethics Committee of the Hospital Universitari Vall d’Hebron, approval code: VDH-VAC-2018-01.

## 4. Statistical Analysis

We described the differences between cases and controls and vaccinated versus unvaccinated patients in terms of demographic, clinical, and laboratory characteristics using the chi-squared test, Fisher’s exact test, and the Wilcoxon rank sum test, as appropriate. As statistical descriptors, we used the median and interquartile range (IQR) for continuous variables, and frequencies and percentages for categorical variables. All statistical tests were two-tailed, and *p*-values under 0.05 were considered statistically significant.

IVE and its 95% corresponding confidence interval (95%CI) were calculated by subtracting the ratio of the odds of vaccination in cases and controls from one, as a percentage (IVE = (1 − odds ratio (OR)) × 100). A bivariate analysis of IVE was performed for all available variables. Multivariate IVE estimates were calculated using logistic regression, adjusting for all available potential confounders described in the literature (age, sex, presence of at least one chronic condition and influenza vaccination in the previous season) [15,16]. We applied a complete case analysis with no imputations. All statistical analyses were carried out in R software (version 4.2.2) [17].

## 5. Results

The DRIVE 2021/22 influenza period in Spain started at epidemiological week (W) 40 of 2021 and ended at W20 of 2022 [18]. At HUVH, the first and last SARI patients within the season were admitted on 29 October 2021 (W43) and 20 April 2022 (W16), respectively. The distribution of cases and controls according to the epidemiological week is depicted in Figure 2. Two epidemiological peaks of SARI were observed, the first between W49 and W50, and the second between W05 and W06. An increase in flu cases was observed between W06 and W14. A total of 260 patients admitted to the hospital met our study criteria (Table 1). The ages ranged from 19 to 96 years and most patients (63%) were older than 65 years of age. A total of 135 participants (58%) were never smokers, while 240 (92%) had at least one chronic condition, predominantly cardiovascular disease (112 patients [43%]), lung disease (98 patients [38%]) and cancer (76 patients [29%]). Additionally, there were 24 patients (9.2%) with a history of immunodeficiency or organ transplantation and 152 participants (58%) had been diagnosed with SARS-CoV-2 infection in the previous 12 months. A total of 25 patients (9.5%) died during admission.

A total of 34 hospitalized patients (13%) tested positive for influenza A virus, all A(H3N2). There were differences between the cases and controls in terms of age (mean = 54.4 [SD = 21.2] vs. 69.1 [13.8], respectively; *p* < 0.001), presence of chronic conditions (81% vs. 93%, *p* = 0.048), and lower pneumococcal vaccination rates (36% vs. 57% *p* = 0.026). Influenza cases had much lower rates of COVID-19 infection in the last 12 months than the controls (21% vs. 64% *p* < 0.001).

The influenza vaccination coverage was 53% (95% CI: 47% to 60%) for all participants. Patients with at least one chronic condition had a vaccination coverage rate of 57% (95% CI: 50% to 63%), while those over 65 years old had a coverage rate of 68% (95% CI: 62% to 67%). There were differences between vaccinated and unvaccinated groups in terms of age (median = 74 [IQR = 66–81] vs. 62 [IQR = 50–73], respectively; *p* < 0.001), history of cardiovascular disease (49% vs. 36%, *p* = 0.041), and cancer (36% vs. 21%, *p* = 0.01). The vaccinated group had significantly higher vaccination coverage rates for all other vaccines considered (Table 2). This included influenza vaccination in the previous season (85% vs. 15%, *p* < 0.001), pneumococcal vaccination (77% vs. 28%, *p* < 0.001), and third dose of COVID-19 vaccination (80% vs. 37%, *p* < 0.001).

### Vaccine Effectiveness

Crude IVE was 36% (95% CI: −34% to 68%) (Table 3). In the bivariate analysis, no significant differences were observed for the studied variables (sex, age, cardiovascular, lung, renal, liver or diabetes disease, immunodeficiency, death, pneumococcal vaccination, COVID-19 infection, and COVID-19 vaccination). The adjusted IVE was 26.4% (95% CI: −69% to 112%).

## 6. Discussion

In this test-negative design (TND) case-control study we report an adjusted IVE for the prevention of hospitalization with SARI due to influenza of 26.4% (95% CI: −69% to 112%) in a third-level hospital in Barcelona, Spain. During the 2021/22 influenza season, we observed a low incidence of hospitalized patients with SARI due to the influenza virus, as compared to previous seasons [9], and all cases observed were caused by the A(H3N2) subtype. The vaccination coverage rate of patients hospitalized due to SARI was 68% (95% CI: 62% to 67%) for those older than 65 years, and 57% (95% CI: 50% to 63%) among patients with at least one chronic condition.

The epidemiological findings of this study are consistent with those observed by national and international surveillance networks that showed a low circulation of influenza, initially superimposed on the SARS-CoV-2 Omicron wave [4,5]. The predominant influenza virus strain throughout the season was A(H3N2). The genetic characterization studies show that the circulating influenza A(H3N2) strain belonged mainly to clade 3C.2a1b.2a.2, which presents changes in the antigenic sites compared to the vaccine strain [19].

In Spain, influenza vaccination is recommended, financed, and promoted for adults aged over 65, individuals with a range of chronic medical conditions, pregnant women, and for occupational sectors that may predispose them to an elevated risk of exposure [20]. Vaccination rates show high variability between regions. In Catalonia, the coverage rate for the 2021/22 season reached 64.1% among adults aged over 65 [14]. In relation to this specific age-related risk group, our study shows similar coverage rates (68%) (95% CI: 62% to 67%). Notably, our findings underscore a lower coverage rate of 57% (95% CI: 50% to 63%) among patients with at least one chronic condition, falling significantly short of the Ministry of Health’s target of 75% [20].

We have observed a marked difference in the history of COVID-19 infections between cases and controls, with cases showing significantly lower infection rates compared to controls (21% vs. 64%, *p* < 0.001). However, we believe that this disparity can be explained mainly because, during the study period, COVID-19 infections were responsible for the majority of SARI hospital admissions; therefore, most of our controls had a history of COVID-19 infection in the last 12 months since they were hospitalized due to COVID-19 infections. Moreover, asymptomatic or unregistered COVID-19 infections may have been more probable among cases as they were significantly younger than controls in our study [21].

For the 2021/22 season, only a single study of IVE in hospitalized patients in the Northern Hemisphere has been published. This study, from the National Influenza Center of the Statens Serum Institute in Denmark, identified 1383 SARIs, also with a high prevalence of H3N2, and a resulting IVE against hospitalization in patients aged over 44 years of −23% (95% CI −44% to −4%) [22]. The rest of the IVE studies published for the 2021/22 season are based on influenza surveillance systems and estimate IVE against influenza-like illness (ILI). In Canada, only 42 cases of H2N3 were identified, with an IVE of 36% (95% CI −38% to 71%) [23]. In the USA, the CDC published very similar results, with an adjusted IVE of 14% (95% CI −17% to 37%) [24]. In Navarra (Spain), an IVE against ILI of 36% (95% CI: 13% to 56%) was estimated [25].

Regarding the studies that focus on IVE against hospitalization, the US Hospitalized Adult Influenza Vaccine Effectiveness Network (HAIVEN) from three seasons (2015–2018) estimated an IVE against hospitalization of 38% (95% CI: 17% to 53%) [26]. A study that is similar to the present one was carried out in Valencia during the 2017/18 season, and obtained a low and non-significant total IVE of 9.9% (95% CI: −15.5% to 29.6%) among patients over 60 years of age, which was highly variable according to the subtypes of influenza virus ranging from −29.88% to 48.33% [27].

Since HUVH is a third-level hospital and a referral centre for highly complex cases, we believe that the present study is useful for estimating IVE in high-risk populations. Conducting IVE studies at the hospital level allows evaluation of the impact of the influenza vaccine in the subgroup of the population that would probably benefit most from it (the elderly and those with complex comorbidities). However, the challenges of estimating IVE are numerous and complex [28]. Observational studies are subject to at least three forms of bias: confounding, selection bias, and information bias. Confounding is caused by the different distribution of risk factors associated with hospitalization among the vaccinated and unvaccinated groups. We have minimized this risk of bias adjusting for the different confounding factors described in the literature. However, residual confounding cannot be ruled out, and has been described in TND studies [29]. A selection bias related to medical care-seeking is controlled by the study design, since both cases and controls had been hospitalized [16]. Information bias was minimized with the requirement of a vaccination registered in the electronic clinical records system shared by all public healthcare centres. Over the last few years, mostly due to the COVID-19 pandemic, a considerable effort in improving the vaccine registers in Catalonia has been made. An important number of studies have been conducted using this source of information [30].

The major limitation of our study is related to the fact that the 2021/22 season was different from the traditional seasonal pattern of influenza, with a marked reduction in the influenza virus circulation [5], probably related to the COVID-19 pandemic non-pharmacologic preventive measures (which reduced virus transmission). The detection of a single circulating strain has hampered the possibility of evaluating the IVE against other strains. However, the primary consequence of the low influenza case count during the 21/22 season is that we were unable to reach our intended sample size. As a result, the effectiveness of the vaccine has been determined with a high degree of uncertainty. Another significant limitation is that the insufficient number of patients obtained for each risk factor, coupled with the marked heterogeneity of the sample, prevents a proper adjustment of the IVE for potential confounders.

## 7. Conclusions

The results of the present study suggest that the effectiveness of the 2021/22 influenza season vaccine in preventing hospitalization are moderate but also highly uncertain, due to the low circulation of the influenza virus. Maintaining surveillance systems for respiratory infections is key to monitoring and evaluating influenza vaccines and consequently facilitates future decision making regarding the use of vaccines in public health efforts.

## Figures and Tables

**Figure 1 vaccines-11-01450-f001:**
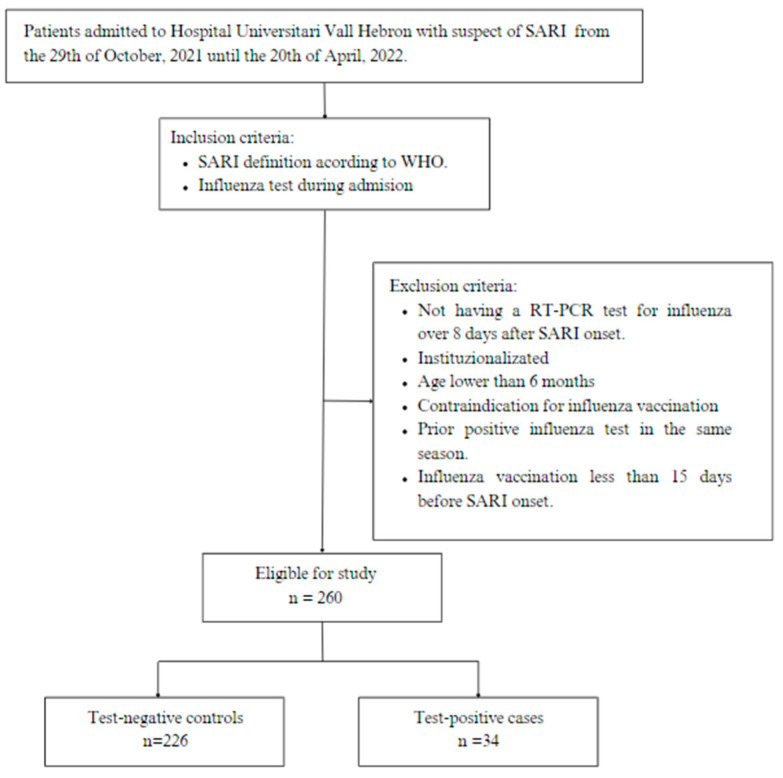
Methodological flowchart: test-negative design (TND) case-control study, Hospital Universitari Vall d’Hebron, 2021/22 influenza season. SARI: Severe Acute Respiratory Infection. WHO: World Health Organization. RT-PCR: Reverse Transcription Polymerase Chain Reaction.

**Figure 2 vaccines-11-01450-f002:**
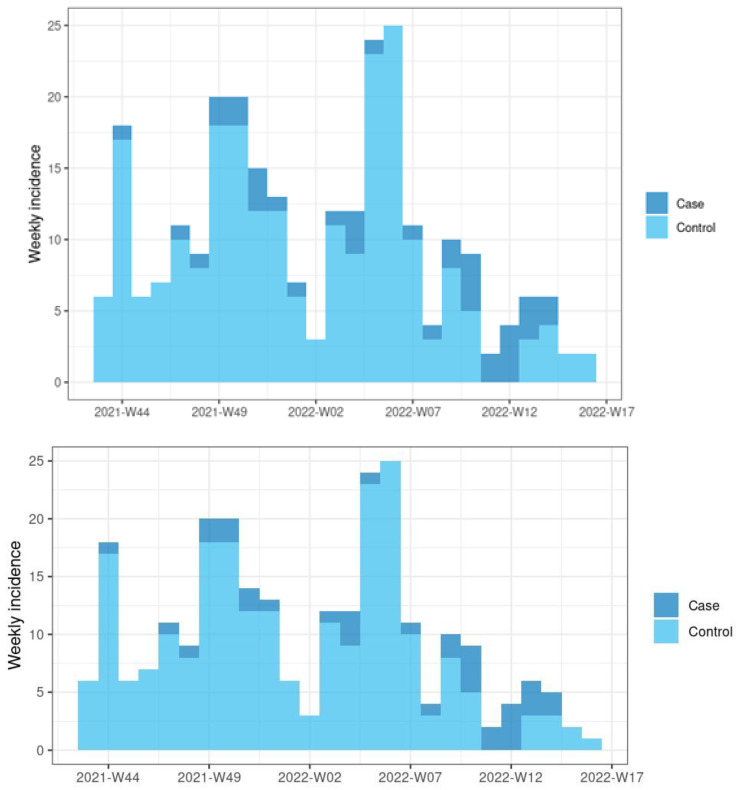
Patients admitted with severe acute respiratory infections during the 2021/22 influenza season, according to epidemiological week, and positive (cases) or negative (controls) influenza RT-PCR test result. Hospital Universitari Vall d’Hebron.

**Table 1 vaccines-11-01450-t001:** Distribution of patients admitted with severe acute respiratory infection. Cases, controls, and vaccinated and unvaccinated according to demographic and clinical characteristics, Hospital Universitari Vall d’Hebron, 2021/22 influenza season.

Characteristic	*n* = 260	Vaccinated *n* = 139	Unvaccinated *n* = 121	*p*-Value ^1,2^	Case *n* = 34	Control *n* = 226	*p*-Value ^1,3^
Age in years: median (IQR)	69 (60–78)	74 (66–81)	62 (50–73)	**<0.001**	59 (37–71)	70 (61–79)	**<0.001**
Age group: <18 y	0/260 (0%)	0/139 (0%)	0/121 (0%)	**<0.001**	0/34 (0%)	0/226 (0%)	**0.021**
Age group: 18–64 y	95/260 (37%)	27/139 (19%)	68/121 (56%)		19/34 (56%)	76/226 (34%)	
Age group: 65+ y	165/260 (63%)	112/139 (81%)	53/121 (44%)		15/34 (44%)	150/226 (66%)	
Sex female	108/260 (42%)	60/139 (43%)	48/121 (40%)	0.6	15/34 (44%)	93/226 (41%)	0.7
Any chronic conditions	238/260 (92%)	135/139 (97%)	103/121 (85%)	**<0.001**	28/34 (82%)	210/226 (93%)	**0.050**
Smoking habits				0.078			0.8
Never	133/232 (57%)	71/129 (55%)	62/103 (60%)		18/34 (53%)	115/198 (58%)	
Ex-smoker	78/232 (34%)	50/129 (39%)	28/103 (27%)		14/34 (41%)	64/198 (32%)	
Occasional	3/232 (1.3%)	2/129 (1.6%)	1/103 (1.0%)		0/34 (0%)	3/198 (1.5%)	
Daily	18/232 (7.8%)	6/129 (4.7%)	12/103 (12%)		2/34 (5.9%)	16/198 (8.1%)	
Unknown	28/260 (11%)	10/139 (7.2%)	18/121 (15%)		0/34 (0%)	28/226 (12%)	
Lung disease	98/260 (38%)	61/139 (44%)	37/121 (31%)	**0.027**	12/34 (35%)	86/226 (38%)	0.8
Renal disease	57/260 (22%)	35/139 (25%)	22/121 (18%)	0.2	9/34 (26%)	48/226 (21%)	0.5
Liver disease	39/260 (15%)	24/139 (17%)	15/121 (12%)	0.3	8/34 (24%)	31/226 (14%)	0.14
Diabetes	64/260 (25%)	39/139 (28%)	25/121 (21%)	0.2	6/34 (18%)	58/226 (26%)	0.3
Cardiovascular disease	112/260 (43%)	68/139 (49%)	44/121 (36%)	**0.041**	13/34 (38%)	99/226 (44%)	0.5
Cancer	76/260 (29%)	50/139 (36%)	26/121 (21%)	**0.010**	9/34 (26%)	67/226 (30%)	0.7
Immunodeficiency or OT	24/260 (9.2%)	13/139 (9.4%)	11/121 (9.1%)	0.9	4/34 (12%)	20/226 (8.8%)	0.5
COVID-19 infection in the past 12 months	150/260 (58%)	77/139 (55%)	73/121 (60%)	0.4	7/34 (21%)	143/226 (63%)	**<0.001**
Death	25/260 (9.6%)	14/139 (10%)	11/121 (9.1%)	0.8	3/34 (8.8%)	22/226 (9.7%)	0.9

^1^ Wilcoxon rank sum test; Fisher′s exact test; Pearson′s chi-squared test. ^2^ Vaccinated vs. unvaccinated *p*-value. ^3^ Influenza cases vs. controls *p*-values. SD: Standard deviation. OT: Organ transplant.

**Table 2 vaccines-11-01450-t002:** Distribution of patients admitted with severe acute respiratory infections. Cases, controls, and vaccinated and unvaccinated according to previous vaccination characteristics, Hospital Universitari Vall d’Hebron, 2021/22 influenza season.

Characteristic	*n* = 260	Vaccinated *n* = 139	Unvaccinated ^1^ *n* = 121	*p*-Value ^1,2^	Case *n* = 34	Control *n* = 226	*p*-Value ^1,3^
Previous vaccination (−1 season)	129/226 (57%)	115/135 (85%)	14/91 (15%)	<0.001	13/26 (50%)	116/200 (58%)	0.4
Unknown	34/260 (13%)	4/139 (2.9%)	30/121 (25%)		8/34 (24%)	26/226 (11%)	
Pneumococcal vaccination	140/259 (54%)	107/139 (77%)	33/120 (28%)	<0.001	12/33 (36%)	128/226 (57%)	0.029
Unknown	1/260 (0.4%)	0/139 (0%)	1/121 (0.8%)		1/34 (2.9%)	0/226 (0%)	
1st COVID-19 vaccine	235/260 (90%)	138/139 (99%)	97/121 (80%)	<0.001	31/34 (91%)	204/226 (90%)	0.9
2nd COVID-19 vaccine	229/260 (88%)	137/139 (99%)	92/121 (76%)	<0.001	31/34 (91%)	198/226 (88%)	0.8
3rd COVID-19 vaccine	156/260 (60%)	111/139 (80%)	45/121 (37%)	<0.001	24/34 (71%)	132/226 (58%)	0.2

^1^ Fisher′s exact test; Pearson′s chi-squared test. ^2^ Vaccinated vs. unvaccinated *p*-value. ^3^ Influenza cases vs. controls *p*-value. SD: Standard deviation. OT: Organ transplant.

**Table 3 vaccines-11-01450-t003:** Crude and adjusted effectiveness of influenza 2021/22 vaccine in preventing hospitalization with laboratory-confirmed influenza, Hospital Universitari Vall d’Hebron.

	Vaccinated *n* = 139	Unvaccinated *n* = 121	Crude Influenza Vaccine Effectiveness (95% CI)	Adjusted Influenza Vaccine Effectiveness * (95% CI)
Case, *n* = 34	15	19	36%	26%
Control, *n* = 226	124	102	(−34% to 68%)	(−69% to 112%)

* Age, sex, presence of at least one chronic condition and influenza vaccination in the previous season.

## Data Availability

The data that support the findings of this study are available from the corresponding author upon reasonable request.

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
