# Peer review of "Influenza Vaccine Effectiveness against Hospitalization, Season 2021/22: A Test-Negative Design Study in Barcelona"

_vaccines, 2023, doi:10.3390/vaccines11091450_

Round 1
Reviewer 1 Report
The authors aim to estimate the IVE against hospitalization and its determinants in a Spanish tertiary hospital during the session in the 2021-22. Hospitalized patients with severe acute respiratory infection (SARI) and available influenza RT-PCR were selected and classified as cases (positive influenza RT-PCR) or controls (negative influenza RT-PCR). The IVE was calculated by subtracting the ratio of the odds of vaccination in cases and controls from one. The SARI were identified, of which 34 were positive for influenza, and all were subtype A(H3N2). 53% were vaccinated. They concluded that the considered data revealed non-significant moderate vaccine effectiveness against hospitalization. Paper is well written, however, we have following concerns:
1) Title is lengthy and needs to be revised.
2) Depict a flowchart (methodological pipeline) to present the whole methodology of system design.
3) The methodology part is very concisely written and hard to follow. It needs detailed description.
4) Add a conclusion section to conclude the paper.
5) The discussion part is not sufficient for the study.
6) Explain properly the importance of each parameter presented in all Tables, in context with the study findings. Like p-value 1 are coming two times in table 1 and 2. Explain what means first p-value 1 and what means second p-value 1 .
6) Paper needs proof reading for English language.
Author Response
We express our gratitude for the feedback you have provided.
- Title is lengthy and needs to be revised.
Thank you for your comment. We have tried to reduce its length as much as possible, leaving all those elements that we consider of value, such as the season, the methodology and the location, in addition to the objective and main result of the study: The effectiveness of the influenza vaccine against hospitalization.
After the editing process, it remains as follows:
Title, page 1:
“Influenza vaccine effectiveness against hospitalization, season 2021/22: a test-negative design study in Barcelona.”
- Depict a flowchart (methodological pipeline) to present the whole methodology of system design.
Thank you for your comment. We addthe flowchart that we have designed, we believe it will be clarifying.
- Methods , 2.1. Study design and population, Page 3:
We conducted a test-negative design (TND) case-control study based on the DRIVE protocol [11.Figure 1 provides a summarized representation of the methodological design in the form of a flowchart.
Figure 1. Methodological Flowchart: test-negative design (TND) case-control study , Hospital Universitari Vall d’Hebron, 2021/22 influenza season.
- The methodology part is very concisely written and hard to follow. It needs detailed description.
We have modified the methodology section with an extension of the definition for some of the variables, in addition to adding the flowchart recommended in the previous comment. We believe that with these changes the text will be more comprehensive. Below, you can see the changes made in the Methods section, highlighted for reference:
Methods, page 2:
- Methods
2.1. Study design and population
We conducted a test-negative design (TND) case-control study based on the DRIVE protocol [11]. Figure 1 provides a summarized representation of the methodological design in the form of a flowchart. It was conducted in Hospital Universitari Vall d’Hebron, a tertiary hospital which provides healthcare coverage to 430,000 inhabitants of Catalonia and is an international referent for highly complex tertiary procedures. The study was conducted during the 2021/22 influenza season, from the 29th of October, 2021 until the 20th of April, 2022.
The study population consisted of all patients admitted to the hospital with SARI according to the WHO accepted definition. This is an acute respiratory infection with a history of fever and cough with onset within last 10 days that requires hospitalization [12]. Following routine procedures, respiratory specimens were collected from all hospitalized SARI patients and tested for influenza using reverse transcription polymerase chain reaction (RT-PCR). The inclusion criteria were: being hospitalized for SARI and having a diagnostic test for influenza performed during the admission. The exclusion criteria were: Being institutionalized, an age lower than 6 months, contraindication for influenza vaccination, prior positive influenza test in the same season, swab taken over 8 days after SARI onset, and influenza vaccination less than 15 days before SARI onset. Patients were classified into cases and controls according to the positive or negative swab test result, respectively.
Based on the hospital’s Electronic Clinical Records, hospitalized patients with signs or symptoms at admission suggestive of SARI were automatically selected. Subsequently, the admission reports were manually reviewed to confirm fulfilment of inclusion criteria.
2.2. Sample size
To calculate the sample size, we used the World Health Organization sample size calculator [13]. Since IVE estimates have wide confidence intervals, we have calculated the sample size using the CIs values obtained from the ECDC metanalysis 44% to 66%. [7], a precision of 40% and an expected vaccination coverage rate of 50% [14]. Therefore, our sample size calculation ranges from 115 to 262 cases and from 115 to 262 controls.
2.3. Data and sources of information
For each patient, demographic information (age and sex), risk factors (smoking habits and comorbilities), as well as, laboratory, and vaccination data were systematically collected from the Hospital Universitari Vall d’Hebron Electonic Clinical Records.
Smoking habits were categorized into 4 groups: never smokers, ex-smokers, occasional smokers, and daily smokers. The following selected comorbidities were collected as cathegorical dichotomous variables according to ICD-10 diagnostic codes: cardiovascular disease, lung disease, renal disease, liver disease, diabetes, cancer and immunodeficency. The variable “any chronic condition” was created taking into account all comorbidities. Laboratory data included influenza type (A and B) and subtype/lineage. Vaccination variables included date of vaccination, type of vaccine and previous vaccination for influenza (only season 20/21), COVID and pneumococcal vaccination record (any time). They were obtained from Història Clínica Compartida de Catalunya (HC3), an Electronic Clinical Records system shared by all public healthcare centers in Catalonia. The variable COVID-19 infection history was defined as RT-PCR or antigen positive SARS-CoV-2 tests in the past 12 months that had been registered in HC3. We did not have access to serological SARS-CoV-2 tests.
The extraction of variables available in the automated hospital records was automatized except for the date of the onset of symptoms, smoking habits, and contraindication for influenza vaccine, which were manually retrieved.
Potencial counfunders or effect modifiers were selected based on population-based vaccine effectiveness studies avaliable in the literature.
Patients with damaged samples, pending laboratory results or missing data on the
laboratory results outcome or covariates were excluded from the analysis.
- Add a conclusion section to conclude the paper.
Following your recommendations, we have added a final section of conclusions.
Conclusion, page 8:
- Conclusion
The results of the present study suggest that the effectiveness of the 2021/22 influenza season vaccine in preventing hospitalization are moderate but also highly uncertain, due to the low circulation of the influenza virus. Maintaining surveillance systems for respiratory infections is key to monitoring and evaluating influenza vaccines and consequently facilitates future decision-making regarding the use of vaccines in public health.
- The discussion part is not sufficient for the study.
We have diligently adhered to the STROBE guidelines, fulfilling all their requirements. We have made modifications in the discussion, specifically, we have refined the sections related to sample size and vaccination rates. Furthermore, we have introduced new content that explores the disparities in the history of COVID infection identified between the cases and controls.
Discussion, page 7:
- Discussion
In this test-negative design (TND) case-control study we report an adjusted IVE for the prevention of hospitalization with SARI due to influenza of 26.4% (95% CI -69% to 112%) in a third level hospital in Barcelona, Spain. During the 2021/22 influenza season we observed a low incidence of hospitalized patients with SARI due to the influenza virus, as compared to previous seasons [9], and all cases observed caused by the A(H3N2) subtype. The vaccination coverage rate of patients hospitalized due to SARI was68% (95% CI: 62% to 67%) for those older than 65 years and 57% (95% CI: 50% to 63%) among patients with at least one chronic condition.
The epidemiological findings of this study are consistent with those observed by national and international surveillance networks that showed a low circulation of influenza, initially superimposed on the SARS-CoV-2 Omicron wave [4][5]. The predominant influenza virus strain throughout the season was A(H3N2). The genetic characterization studies show that the circulating influenza A(H3N2) strain belonged mainly to clade 3C.2a1b.2a.2, which presents changes in the antigenic sites compared to the vaccine strain [19].
In Spain, influenza vaccination is recommended, financed and promoted for adults aged over 65, individuals with a range of chronic medical conditions, pregnant women and for occupational sectors that may predispose them to an elevated risk of exposure [20]. Vaccination rates show high variability between regions. In Catalonia, the coverage rate for the 2021/22 season reached 64.1% among adults aged over 65 [14]. In relation to this specific age-related risk group, our study shows similar coverage rates (68%) (95% CI: 62% to 67%). Notably, our findings underscore a lower coverage rate of 57% (95% CI: 50% to 63%) among patients with at least one chronic condition, falling significantly short of the Ministry of Health's target of 75% [20].
We have observed a marked difference in the history of COVID infections between cases and controls, with cases showing significantly lower infection rates compared to controls (21% vs 64%, p<0.001). However, we believe that this disparity can be explained mainly because during the study period COVID-19 infections were responsible for the majority of SARI hospital admissions, therefore, most of our controls had a history of COVID infection in the last 12 months since they were hospitalized due to COVID-19 infections. Moreover, asymptomatic or unregistered COVID-19 infections may have been more probable among cases since cases where significantly younger than controls in our study [21].
For the 2021/22 season, only a single study of IVE in hospitalized patients in the Northern Hemisphere has been published. This study, from the National Influenza Center of the Statens Serum Institute in Denmark, identified 1383 SARIs, also with a high prevalence of H3N2, and a resulting IVE against hospitalization in patients aged over 44 years of -23% (95% CI -44% to -4%) [22]. The rest of the IVE studies published for the 2021/22 season are based on influenza surveillance systems and estimate IVE against influenza-like illness (ILI). In Canada, only 42 cases of H2N3 were identified, with an IVE of 36% (95% CI -38% to 71%) [23]. In the USA, the CDC published very similar results, with an adjusted IVE of 14% (95% CI -17% to 37%) [24]. In Navarra (Spain) an IVE against ILI of 36% (95% CI: 13% to 56%) was estimated [25].
Regarding the studies that focus on IVE against hospitalization, the US Hospitalized Adult Influenza Vaccine Effectiveness Network (HAIVEN) from 3 seasons (2015-2018) estimated an IVE against hospitalization of 38% (95% CI 17% to 53%) [26]. A study that is similar to the present one was carried out in Valencia during the 2017/18 season, and obtained a low and non-significant total IVE of 9.9% (95% CI: -15.5% to 29.6%) among patients over 60 years of age, which was highly variable according to the subtypes of influenza virus ranging from −29.88% to 48.33% [27].
Since HUVH is a third level hospital and a referral centre for highly complex cases, we believe that the present study is useful for estimating IVE in high-risk populations. Conducting IVE studies at the hospital level allows to evaluate the impact of the influenza vaccine in the subgroup of the population that would probably benefit most from it (the elderly and those with complex comorbidity). However, the challenges of estimating IVE are numerous and complex [28]. Observational studies are subject to at least three forms of bias: confounding, selection bias, and information bias. Confounding is caused by the different distribution of risk factors associated with hospitalization among the vaccinated and unvaccinated groups. We have minimized this risk of bias adjusting for the different confounding factors described in literature. However, residual confunding cannot be ruled out, and has been described in TND studies [29]. A selection bias related to medical care-seeking is controlled by the study design, since both cases and controls had been hospitalized [16]. Information bias was minimized with the requirement of a vaccination registered in the Electronic Clinical Records system shared by all public healthcare centres Over the last years, mostly due to the COVID pandemic, a great effort in improving the vaccine registers in Catalonia has been made. An important number of studies have been conducted using this source of information [30].
The major limitation of our study is related to the fact that the 2021/22 season was different from the traditional seasonal pattern of influenza with a marked reduction in the influenza virus circulation [5], probably related to the COVID-19 pandemic non-pharmacologic preventive measures (which reduced virus transmission). The detection of a single circulating strain has hampered the possibility of evaluating the IVE against other strains. However, the primary consequence of the low influenza case count during the 21/22 season is that we were unable to reach our intended sample size. As a result, the effectiveness of the vaccine has been determined with a high degree of uncertainty. Another significant limitation is that the insufficient number of patients obtained for each risk factor, coupled with the marked heterogeneity of the sample, prevents a proper adjustment of the IVE for potential confounders.
- Explain properly the importance of each parameter presented in all Tables, in context with the study findings. Like p-value 1 are coming two times in table 1 and 2. Explain what means first p-value 1 and what means second p-value 1.
Thank you for your comment. We have added a superscript specifying which columns are being compared to obtain the p-value.
Results, Table 1:
2 Vaccinated vs Unvaccinated p-value.
3 Influenza Cases vs Controls p-value.
- Paper needs proof reading for English language.
Thank you for your comment. We would like to highlight that the manuscript has undergone language revision conducted by an external reviewer with native-level proficiency in English, along with the requisite certification. The reviewer, Elfriede Laaff, is duly acknowledged in the paper's acknowledgments section.
Reviewer 2 Report
I have found the manuscript well written and presented. Data from a single hospital and a single flu season have not enough potential to get significant results. Surveillance networks and other research consortiums are stablished to have representative data available and then more robust and significant results.
Nevertheless, the strenghts of your work - the methodology, the design and the analisys- deserve to be published
Points to be revised:
The references must be revised, because they are displaced. (eg the Reference 5 in the text correspond to reference 4 in the list of references...);
Reference 7 is published in 2012 I find is not “recent”, please revise the text or update the reference
Table 2: the rows with the number of registries with unknown information seem strange …..could you propose an other way to show this information
ECR in mentioned firstly in the Discussion, please explain the acronymus and add more information about the system .
Author Response
We express our gratitude for the feedback you have provided.
Points to be revised:
Thank you very much for your comments. We truly appreciate your support in advocating for the promotion of our study's publication. Concerning the points to be revised:
The references must be revised, because they are displaced. (eg the Reference 5 in the text correspond to reference 4 in the list of references...);
Reference 7 is published in 2012 I find is not “recent”, please revise the text or update the reference
We have identified an error in the formatting where numbering started at the 2nd reference, causing a misalignment with a +1 discrepancy throughout. I have diligently reviewed them, and they now align. With the correction you can see that reference 7 is from 2020.
Table 2: the rows with the number of registries with unknown information seem strange …..could you propose an other way to show this information
Registries with unknown information: In an effort to enhance clarity regarding the missing data, we have incorporated the denominator representing the total for each group with unknown information. We think this also helps in compressing that frequency, since it doesn't account for missing data. Below, we present the table where the changes have been highlighted. We have extended the modification to table 1.
|
Characteristic |
N = 260 |
Vaccinated N = 139 |
Unvaccinated1 N = 121 |
p-value1,2 |
Case N = 34 |
Control N = 226 |
p-value1,3 |
|
Previous vaccination (-1 season) |
129 / 226 (57%) |
115 / 135 (85%) |
14 / 91 (15%) |
<0.001 |
13 / 26 (50%) |
116 / 200 (58%) |
0.4 |
|
Unknown |
34 / 260 (13%) |
4 / 139 (2.9%) |
30 / 121 (25%) |
|
8 / 34 (24%) |
26 / 226 (11%) |
|
|
Pneumococcal vaccination |
140 / 259 (54%) |
107 / 139 (77%) |
33 / 120 (28%) |
<0.001 |
12 / 33 (36%) |
128 / 226 (57%) |
0.029 |
|
Unknown |
1 / 260 (0.4%) |
0 / 139 (0%) |
1 / 121 (0.8%) |
|
1 / 34 (2.9%) |
0 / 226 (0%) |
|
|
1st COVID vaccine |
235 / 260 (90%) |
138 / 139 (99%) |
97 / 121 (80%) |
<0.001 |
31 / 34 (91%) |
204 / 226 (90%) |
0.9 |
|
2nd COVID vaccine |
229 / 260 (88%) |
137 / 139 (99%) |
92 / 121 (76%) |
<0.001 |
31 / 34 (91%) |
198 / 226 (88%) |
0.8 |
|
3rd COVID vaccine |
156 / 260 (60%) |
111 / 139 (80%) |
45 / 121 (37%) |
<0.001 |
24 / 34 (71%) |
132 / 226 (58%) |
0.2 |
|
1 Fisher's exact test; Pearson's Chi-squared test 2 Vaccinated vs Unvaccinated p-value. 3 Influenza Cases vs Controls p-value. SD: Standard deviation OT: Organ Transplant |
|||||||
ECR in mentioned firstly in the Discussion, please explain the acronymus and add more information about the system.
The acronym was introduced in the Methods section. However, we have opted to remove the acronym to improve the readability of the paper. Thank you for your feedback
Reviewer 3 Report
This is a well written and interesting - given that flu and fly vaccination affects millions - ms. While the authors did nor reach their target sample size by a wide margin (which the authors clearly admit) this is unavoidable and the authors' conclusion is sufficiently carefully formulated.
I have but two minor comments.
Where a history of covid infection is compared, it is essential to state how this was established. The age difference between cases and control makes it more likely that cases experienced sub-clinical infection. Such difference is less likely if infection was established by serology.
"residual confusion"? I presume "residual confounding" is what is intended?
Author Response
We express our gratitude for the feedback you have provided.
Thank you very much for your comments. As you rightly point out, we are fully aware that our result lacks significance due to the limited sample size of cases, further diminished by the unique characteristics of the 2021/22 flu season, characterized by extremely low flu circulation. Nonetheless, we firmly believe that the strengths of our work - the methodology, design, and analysis - warrant publication.
I have but two minor comments.
History of covid
Where a history of covid infection is compared, it is essential to state how this was established. The age difference between cases and control makes it more likely that cases experienced sub-clinical infection. Such difference is less likely if infection was established by serology.
The history of COVID infection was extracted from regional electronic clinical records containing virological diagnostic tests (Antigen detection or RT-PCR) conducted within public healthcare services. Serology was never utilized for this purpose. Hence, we agree with your observation that cases of influenza, younger than controls, probably have experienced asymptomatic and thus, undiagnosed and unrecorded, COVID infections. We also acknowledge the potential overrepresentation of COVID infections in the control group, given that during the 2021/22 season, COVID infections caused the majority of hospital admissions for severe respiratory infections, which was counted as an infection in the last 12 months. Besides, there might be an underestimation in cases due to registration errors.
To address potential confusion stemming from your feedback, we have included a sentence in the Methods section to elucidate this variable and also everything previously mentioned in the discussion. The added sentences are provided below:
Methods, Data and sources of information, page 4:
The variable COVID-19 infection history was defined as RT-PCR or antigen positive SARS-CoV-2 tests in the past 12 months that had been registered in HC3. We did not have access to serological SARS-CoV-2 tests.
Discussion, page 8:
We have observed a marked difference in the history of COVID infections between cases and controls, with cases showing significantly lower infection rates compared to controls (21% vs 64%, p<0.001). However, we believe that this disparity can be explained mainly because during the study period COVID-19 infections were responsible for the majority of SARI hospital admissions, therefore, most of our controls had a history of COVID infection in the last 12 months since they were hospitalized due to COVID-19 infections. Moreover, asymptomatic or unregistered COVID-19 infections may have been more probable among cases since cases where significantly younger than controls in our study [21].
Residual confusion
"residual confusion"? I presume "residual confounding" is what is intended?
You are absolutely right, and we appreciate the correction. The language error has been rectified
Discussion, page 8:
We have minimized this risk of bias adjusting for the different confounding factors described in literature. However, residual confunding cannot be ruled out, and has been described in TND studies [29].
Reviewer 4 Report
Introduction
“According to 70 a recent systematic review by the ECDC, trivalent IVE across three influenza seasons was 71 estimated to be 45% (95% CI: 23% to 61%) against infection for any influenza virus type, 72 but 61% (95% CI: 44% to 73%) for influenza A(H1N1)pdm09, 29% (95% CI: 5% to 46%) for 73 influenza B and 11% (95% CI: -25% to 36%) for influenza A(H3N2) [7].”
[These are very wide 95%Cis. How doe these very wide 95% Cis affect the accuracy of your VE computations?]
“The exclusion criteria were: being institutionalized”
[why did you exclude institutionalised patients? They are likely to be more highly susceptible to influenza and have lower influenza vaccine VE’s]
Methods
“To calculate the sample size, we used the World Health Organization sample size 115 calculator [13]. Using a predicted IVE of 60% [7], a precision of 40% and an expected vac- 116 cination coverage rate of 50% [14] we estimated a sample size of 148 cases and 148 controls.”
[This is a very low precision value. Why did you choose this> What sample sizes would have been required for larger precision values?]
“A total of 260 patients admitted to the hospital met our 159 study criteria (Table 1). The ages ranged from 19 to 96 years and most patients (63%) were 160 older than 65 years of age. A total of 135 (58%) of the participants were never smokers, 161 while 240 (92%) had at least one chronic condition, predominantly cardiovascular disease 162 (112 patients [43%]), lung disease (98 patients [38%]) and cancer (76 patients [29%]). Ad- 163 ditionally, there were 24 (9.2%) patients with a history of immunodeficiency or organ 164 transplantation and 152 (58%) participants had been diagnosed of SARS-CoV-2 infection 165 in the previous 12 months. A total of 25 (9.5%) patients died during admission.”
[This a very heterogeneous group of patients, as expected in a hospital of your type. You do not have enough patients with each risk factor are thus not able to control for enough risk factors and you need to point this out please].
Results
“A total of 34 hospitalized patients (13%) tested positive for influenza A virus, all 170 A(H3N2). There were differences between cases and controls in terms of age (mean = 54.4 171 [SD = 21.2] vs. 69.1 [13.8], respectively …The influenza vaccination coverage was 53% for all participants. Patients with at least 176 one chronic condition had a vaccination coverage rate of 57%, while those over 65 years 177 old had a coverage rate of 68%”
[again, do you have enough numbers to draw conclusions?]
Discussion
“The low incidence of influenza has not allowed 204 us to explore the effect of IVE predictors.”
In view of all the problems in your study (which you are not responsible for) the study would be best written up as a shorter research note.
Author Response
We express our gratitude for the feedback you have provided.
Introduction
“According to 70 a recent systematic review by the ECDC, trivalent IVE across three influenza seasons was 71 estimated to be 45% (95% CI: 23% to 61%) against infection for any influenza virus type, 72 but 61% (95% CI: 44% to 73%) for influenza A(H1N1)pdm09, 29% (95% CI: 5% to 46%) for 73 influenza B and 11% (95% CI: -25% to 36%) for influenza A(H3N2) [7].”
[These are very wide 95%Cis. How doe these very wide 95% Cis affect the accuracy of your VE computations?]
Thank you for your question. These confidence intervals correspond to the results of meta-analyses (both fixed-effects and random-effects) published in what they term as a systematic review. We understand that the width of these confidence intervals should not necessarily impact our calculation of the IVE. We have added the clarification that the results stem from a meta-analysis.
Methods, Sample size, pag 3:
To calculate the sample size, we used the World Health Organization sample size calculator [13]. Since IVE estimates have wide confidence intervals, we have calculated the sample size using the CIs values obtained from the ECDC metanalysis 44% to 66%. [7], a precision of 40% and an expected vaccination coverage rate of 50% [14]. Therefore, our sample size calculation ranges from 115 to 262 cases and from 115 to 262 controls.
“The exclusion criteria were: being institutionalized”
[why did you exclude institutionalised patients? They are likely to be more highly susceptible to influenza and have lower influenza vaccine VE’s]
We appreciate your question regarding the exclusion criteria. The decision to exclude institutionalized patients was made based on several factors. Institutionalized patients often present unique health profiles and living conditions that could potentially introduce confounding variables or distinct patterns of disease exposure. By excluding this particular group, we aimed to maintain a more homogenous study population. However, it's important to note that this exclusion was carefully considered and is not meant to generalize to the broader population.
Methods
“To calculate the sample size, we used the World Health Organization sample size 115 calculator [13]. Using a predicted IVE of 60% [7], a precision of 40% and an expected vac- 116 cination coverage rate of 50% [14] we estimated a sample size of 148 cases and 148 controls.”
[This is a very low precision value. Why did you choose this> What sample sizes would have been required for larger precision values?]
Thank you for your comment. A relatively low precision was used in the sample size calculation in accordance with the nature of the study, as it was a single-center study. For 30% precision it would have been required 255 cases and 255 controls, for a 20% precision, 559 cases and 559 controls, for a 10% precision 2200 cases and 2200 controls.
“A total of 260 patients admitted to the hospital met our 159 study criteria (Table 1). The ages ranged from 19 to 96 years and most patients (63%) were 160 older than 65 years of age. A total of 135 (58%) of the participants were never smokers, 161 while 240 (92%) had at least one chronic condition, predominantly cardiovascular disease 162 (112 patients [43%]), lung disease (98 patients [38%]) and cancer (76 patients [29%]). Ad- 163 ditionally, there were 24 (9.2%) patients with a history of immunodeficiency or organ 164 transplantation and 152 (58%) participants had been diagnosed of SARS-CoV-2 infection 165 in the previous 12 months. A total of 25 (9.5%) patients died during admission.”
[This a very heterogeneous group of patients, as expected in a hospital of your type. You do not have enough patients with each risk factor are thus not able to control for enough risk factors and you need to point this out please].
Thank you for your comment. We agree with your reasoning. We have expanded the discussion at this point to provide further clarification:
Discussion, page 8:
“The major limitation of our study is related to the fact that the 2021/22 season was different from the traditional seasonal pattern of influenza with a marked reduction in the influenza virus circulation [5], probably related to the COVID-19 pandemic non-pharmacologic preventive measures (which reduced virus transmission). The detection of a single circulating strain has hampered the possibility of evaluating the IVE against other strains. However, the primary consequence of the low influenza case count during the 21/22 season is that we were unable to reach our intended sample size. As a result, the effectiveness of the vaccine has been determined with a high degree of uncertainty. Another significant limitation is that the insufficient number of patients obtained for each risk factor, coupled with the marked heterogeneity of the sample, prevents a proper adjustment of the IVE for potential confounders.
Results
“A total of 34 hospitalized patients (13%) tested positive for influenza A virus, all 170 A(H3N2). There were differences between cases and controls in terms of age (mean = 54.4 171 [SD = 21.2] vs. 69.1 [13.8], respectively …The influenza vaccination coverage was 53% for all participants. Patients with at least 176 one chronic condition had a vaccination coverage rate of 57%, while those over 65 years 177 old had a coverage rate of 68%”
[again, do you have enough numbers to draw conclusions?]
Thank you for your comment. Regarding the text you provided, it pertains to a description of the sample, we coment diffences observed in it based on statistical test. Concerning vaccination rates, we add their confidence intervals and expanded the discussion section.
Results, pag 6:
The influenza vaccination coverage was 53% (95% CI: 47% to 60%) for all participants. Patients with at least one chronic condition had a vaccination coverage rate of 57% (95% CI: 50% to 63%), while those over 65 years old had a coverage rate of 68% (95% CI: 62% to 67%).
Discusion, pag 8
... In Catalonia, the coverage rate for the 2021/22 season reached 64.1% for among adults aged over 65 [14]. In relation to this specific age-related risk group, our study shows similar coverage rates (68%) (95% CI: 62% to 67%). Notably, our findings underscore a lower coverage rate of 57% (95% CI: 50% to 63%) among patients with at least one chronic condition, falling significantly short of the Ministry of Health's target of 75% [20]
Discussion
“The low incidence of influenza has not allowed 204 us to explore the effect of IVE predictors.”
In view of all the problems in your study (which you are not responsible for) the study would be best written up as a shorter research note.
Thank you for your comment. We will take into consideration all your recommendations and guidance for future research projects.